# Systematic review and narrative synthesis of the experiences of individuals with chronic pain participating in digital pain management interventions

**Justin Damian Russell Strain**[1]*, **Lindsay Welch**[2,3], **Euan Sadler**[4]

**1** Consultant Musculoskeletal Physiotherapist, Southern Health NHS Foundation Trust, Lymington, England, **2** Associate Professor of Nursing Practice, Faculty of Health and Social Sciences, Bournemouth University, Bournemouth, England, **3** University Hospital Dorset NHS Trust, Bournemouth, England, **4** Associate Professor in Social Science, Health & Ageing, School of Health Sciences, University of Southampton, Southampton, England

* justin.strain@southernhealth.nhs.uk

## Abstract

### Background

The use of digital pain management interventions has grown since the Covid 19 pandemic. The aim of this study was to systematically review and synthesise evidence from qualitative studies regarding the experiences of individuals with chronic pain participating in digital pain management interventions in primary care and community settings.

### Methods

Fourteen databases were searched, as well as citation tracking and hand-searching reference lists of included articles. The latest search was completed by 07/07/2023. Qualitative studies of patient and carer perspectives of digital pain management interventions for adults aged 18 and over with non-malignant chronic pain were included. All studies were appraised for quality using the Critical Appraisal Skills Programme Qualitative Checklist. A narrative synthesis approach was used to synthesise the findings. Normalisation Process Theory was used to understand how individuals with chronic pain make sense of digital pain management interventions and incorporate knowledge, skills and strategies learnt into their day-to-day lives.

### Results

Eleven studies, encompassing both digital applications for use on smartphones/ mobile devices and user-directed online modular programmes, were included in the synthesis. Three main themes and related subthemes were identified from the included studies: 1) Making sense of the digital intervention (Subthemes: Tailoring to user's needs; Human contact and support; Accessibility of the digital intervention; Personal and environmental factors affecting engagement with digital interventions); 2) Initiating and Maintaining Behaviour

**Data Availability Statement:** All relevant data are within the manuscript.

**Funding:** JS received a Research Internship and Research Initiation Award to provide funded time

for this research. This included academic supervision from ES and LW. These awards were both funded by NIHR Applied Research Collaboration (ARC) Wessex (https://www.arc-wx. nihr.ac.uk/). ES is supported by NIHR ARC Wessex. The funders had no role in study design, data collection and analysis, decision to publish, or preparation of the manuscript.

**Competing interests:** The authors have declared that no competing interests exist.

Change (Subthemes: Planning activity; Being active); and 3) Personal Growth (Subthemes: Gaining understanding and skills; Gaining and acting on feedback; Negotiating a new relationship with pain).

## Conclusion

**Recommendations**. The key recommendations from our findings are that digital pain management interventions should provide:

- Specific and tailored information for individual participants.

- Focus on changing attitudes and behaviours and reframing perceptions of pain.

- Structured goal setting with prompts to review goals.

- Potential healthcare professional support alongside the digital intervention.

   **Limitations of the review.**   To reduce bias, it would have been preferable for more than one author to independently fully analyse each paper and to identify themes and sub-themes. Instead, the identified themes and sub-themes were discussed with two other authors in the team (ES, LW) to reach a consensus view on final themes and sub-themes. One author (JS) received a Research Internship and Research Initiation Award funded by NIHR Applied Research Collaboration (ARC) Wessex (https://www.arc-wx.nihr.ac.uk/) and NHS England (https://www.england.nhs.uk/). The protocol for this review was registered with the National Institute of Health Research (NIHR) PROSPERO international database for registering systematic reviews (PROSPERO Registration Number CRD42021257768).

## Background

Chronic pain is defined as pain that persists or recurs for more than 12 weeks, despite medication or treatment [1, 2]. It is a long-term condition which costs the UK economy £10 billion per year [3]. The Health Survey for England [4] found that 34% of adult respondents experienced chronic pain. Of those reporting chronic pain, 34% reported this had a significant negative impact on their daily activities.

   It is generally accepted that effective pain management service provision requires a multi-disciplinary team approach including psychosocial and behavioural elements such as managing pain-associated distress and physical activity [5]. The use of digital technology approaches in pain management (including remote consultations via digital platforms, digital group programmes, workshops and modular learning packages) has grown both prior to the COVID 19 crisis [6] and, along with other healthcare interventions, has expanded rapidly during the COVID 19 pandemic within a health service delivery context [7]. This represents a significant shift from usual face-to-face to digital healthcare delivery of pain management interventions. The impact of these recent changes for patients is not fully understood. The use of digital and online health technology in pain management expanded rapidly during the during the COVID 19 crisis, driven by the need to reduce face-to-face contact due to the risk of infection. The urgency of this change meant that reduced time was spent on researching and developing the design and rigorously testing the intervention [8].

   There is currently little evidence about individuals' experiences of digital pain management interventions and the factors perceived to facilitate or hinder their engagement [9]. Given the increasing use of digital pain management interventions, it is important to understand how

digital healthcare technology can be best used to potentially reduce healthcare costs, improve access to healthcare interventions and maximise the effectiveness of future digital pain management interventions [7, 8]. Cronström et al (2019) [10] reported that some individuals with osteoarthritic pain found digital interventions more accessible. However, patients with low back pain in a study exploring patient uptake of digital interventions reported that digital interventions may be more difficult to engage with due to lack of motivation and support, difficulty using information technology, or a perceived lack of personal relevance [11]. Practical considerations such as accessibility and availability of suitable computer equipment could also provide barriers to individuals with chronic pain engaging with online pain management interventions [11].

Research is needed to understand how the shift from face-to-face to digital pain management approaches have shaped experiences, use and perceived benefits of these digital interventions for individuals with chronic pain, including the impact on their capacity to self-manage, physical function and activity levels and how they incorporate knowledge, skills and strategies learnt into their day-to-day lives.

In this study, we drew on the Normalisation Process Theory (NPT) [12–14] as a useful framework to aid interpretation of findings to understand how individuals with chronic pain made sense of and implemented the digital intervention in their daily lives. NPT describes four inter-related core constructs involved in the process of implementing a new innovation into practice or a person's everyday life. Individuals first need to make sense of the new practice or process and understand its relevance to them, within their own social context (*Coherence*). They also need to participate and interact with other people and resources to plan how the innovation will work within their lives (*Cognitive Participation*). Individuals then need to act both individually and collectively with others to further develop the knowledge, skills and experience to put the innovation into practice (*Collective Action*). There is also an ongoing process of reflection and monitoring that enables people to examine the impact of the new innovation and adapt, embed or 'normalise' any changes (e.g. behaviour changes) in the longer term (*Reflexive Monitoring*) [12, 15, 16].

## Aim and objectives

The aim of this study was to systematically review and synthesise evidence from qualitative studies regarding the experiences of individuals with chronic pain participating in digital pain management interventions in primary care and community settings.

### Specific objectives were.

- To examine perceptions of the impact of digital pain management interventions, including their capacity to self-manage, physical function and activity levels among individuals with chronic pain.

- To examine the perceived barriers and facilitators for individuals with chronic pain to engage with digital pain management interventions and use knowledge, skills and strategies learnt in the day-to-day lives.

### Registration

The protocol for this review was registered with the National Institute of Health Research (NIHR) PROSPERO international database for registering systematic reviews (PROSPERO Registration Number CRD42021257768).

## Methods and methodology

### Search strategy

To find and select relevant studies for a qualitative evidence synthesis, a robust search strategy was conducted. One author (JS) used several methods to search for eligible studies, including electronic database searches, citation tracking, recommendations from experts and hand-searching of reference lists. A systematic search of electronic databases was initially conducted by one author (JS) from inception of each database until April 2021. This search was completed by 11/06/2021. A further search of each database from April 2023 until June 2023 was completed by 07/07/2023. Databases searched were CINAHL, Cochrane Library, EMBASE, MEDLINE, PsychInfo, Pub Med, SCOPUS, Web of Science, NICE Website, NHS England Website, NIHR Library, Europe PMC, NHS Digital Website. Grey literature was also searched using the following databases: Health Management Information Centre, King's Fund, Health Foundation, Google, App Store, DARE, and OpenGrey.

The search strategy used free text terminology, combining the terms: [Chronic Pain OR Persistent Pain OR Pain Management] AND [Digital OR Online] AND Qualitative AND [Patients' Experience* OR Patients' Perception*OR Self-Management OR Physical Function OR Physical Activity OR Facilitator* OR Barrier*].

Following identification of articles from electronic databases, the same author (JS) conducted citation tracking in Google Scholar and hand-searching of reference lists of included studies. Further potentially eligible studies were identified through recommendations from experts in the field and by reviewing digital academic profiles of the authors of included papers.

The screening of papers involved a three-stage process. Firstly, titles of all identified articles were screened, and non-relevant studies were excluded, for example, quantitative study designs and studies that were not related to chronic pain. A second screening was then conducted of the remaining abstracts and at this stage any duplicates were removed. Thirdly, potentially eligible studies were obtained and read in full. Articles that did not meet the inclusion criteria (see Table 1) after full-text reading were excluded. Reasons for exclusion at this stage included articles that were not exploring digital interventions, quantitative studies, studies that were not primary research (e.g. systematic reviews), participants under the age of eighteen, and studies that examined health care professionals' experiences.

### Quality appraisal

The quality of included studies was appraised using the Critical Appraisal Skills Programme (CASP) Qualitative Research Checklist [17]. This is a 10-question checklist, designed to assess

**Table 1. Inclusion and exclusion criteria for studies in the review.**

| Inclusion Criteria for Studies | Exclusion Criteria for Studies |
|---|---|
| Service user/patient and carer perspectives of digital pain management interventions. | Health and social care professional/provider perspectives of digital pain management interventions. |
| Older adults aged 18 years and over. | Children (aged under 18 years). |
| Service users/patients with non-malignant chronic pain. | Patients with acute or malignant pain/conditions. |
| Qualitative studies (i.e. interviews, focus groups, ethnographic studies, mixed methods studies with a substantial qualitative component). | Quantitative Studies. |
| Peer reviewed empirical studies. | Studies of face-to-face Pain Management Interventions. |
| Grey Literature, such as support information and patient feedback for currently available digital pain management interventions. | |

methodological quality and validity of findings of qualitative research in which studies are rated as either high, medium or low quality. Two researchers (JS and JH) independently rated the quality of included studies, from the database search completed on 11/06/21, using the CASP checklist and then reached a consensus on the quality rating. One researcher (JS) rated the quality of included studies, from the database search completed on 07/07/23, using the CASP checklist. CASP quality ratings are included in the Characteristics of included studies table (see Table 2).

### Data synthesis, analysis, and assessment of robustness of synthesised themes

A narrative synthesis approach [18] was used to provide a narrative, textual account of the experiences of individuals with chronic pain participating in digital pain management interventions. We used a process of tabulation and thematic analysis to identify initial themes and related sub-themes across included studies. Then we used the 'One Sheet of Paper' (OSOP) method [19] to visually map and synthesise relationships between themes and related sub-themes, noting similarities and differences across included studies. The process was repeated several times to enable the development of 'higher level' synthesised themes across the studies. Final themes were discussed and agreed between three authors (JS, ES, LW) which also involved a group discussion and consensus on how these themes mapped to the four constructs of NPT to aid interpretation of findings.

Assessment of the robustness of the theoretically informed synthesised themes was the final step used in the narrative synthesis. To reduce potential bias, the findings were reanalysed after removing those studies with a medium or low CASP quality rating [20–22]. This reanalysis confirmed the original synthesised themes and subthemes were robust.

## Results

3834 study titles were identified for initial screening against the inclusion and exclusion criteria. Abstracts of 326 articles were retrieved and screened and 57 potentially relevant articles were identified for full-text reading. Following full-text reading, 46 articles were excluded. Reasons for exclusion included quantitative designs; studies focused on healthcare professional perspectives; studies informing development of interventions; and those not focusing specifically on pain management. This left eleven eligible studies. The results from the search strategy are summarised in the PRISMA flow diagram (Fig 1).

### Description of included studies

Eleven studies were included from Australia (N = 2), Denmark (N = 1), England (N = 2), Hong Kong (N = 1), Norway (N = 1), Scotland (N = 1), Sweden (N = 2), and the United States of America (N = 1). The timeframe in which the studies were included ranged from 2009 to 2021. Studies encompassed both digital applications for use on smartphones/ mobile devices [21, 23–25] and user-directed online modular programmes [9, 20, 22, 26–29].

Interventions varied in their levels of human/ healthcare professional contact and support for people with chronic pain (Table 2). All studies were primarily focussed on the self-directed use of web-based applications [9, 20–29]. Five studies [9, 22, 26–28] also included healthcare professional contact during the use of a digital application. One study [23] included brief training by the research team on the use of a digital smartphone application to support older people with arthritis in managing their pain. In two other studies [21, 24] there was no healthcare professional contact but some researcher contact prior to using the self-management intervention [21] or during the period of use [24].

**Table 2. Characteristics of included studies.**

| Authors | Year and Country | Aim of Study | Intervention format | Study Design | Participants | Data Collection | Data Analysis | Main Themes | Quality Appraisal (CASP) |
|---|---|---|---|---|---|---|---|---|---|
| Bendelin et al | 2020 Sweden | To explore how IACT was experienced by participants in a randomised controlled trial of a digital self-directed pain management programme. | Psychology (ACT) based digital self-directed pain management programme delivered to participants in their homes via the internet. Programme involved weekly health professional contact via the internet and telephone contact with a therapist on 2 occasions during the programme. | Qualitative exploration following randomised controlled trial. | 11 adults who had completed an Internet Acceptance and Commitment Therapy (IACT) web-based psychology programme for chronic pain. | Semi-structured interviews | Grounded theory | • Physical and Cognitive Restraints • Deadlines and Time • Therapist Contact • Self-confrontation • Attitudes to Pain • Image of Pain • Control or Command • Acting with Pain • Autonomy • Psychological Flexibility • Treatment Expectations • Therapist Guidance. | High |
| Bhattarai et al | 2020 Australia | To explore attitudes and experiences of older people with chronic arthritic pain to using an app for their pain self-management. | Digital self-directed app for people with Rheumatoid Arthritis, including self-assessment of pain and activity levels and pain self-management education. Participants received brief training in use of the App. | Qualitative study with constructivist worldview. | 18 older adults (age, e.g 60 years and over) who completed 2-week pain app– Rheumatoid Arthritis Information Support and Education (RAISE). | Semi-structured telephone interviews 2 weeks after completion of pain self-management app. | Integrative thematic analysis using both inductive and deductive approaches. | • Apps are a valuable self-management tool but they also have the potential for harm. • A pain self-management app needs to strictly align with the user's needs. • Clinicians' involvement was crucial when integrating an app into older people's pain self-management regime. • Pain self-management apps must be designed with the end-user in mind. | High |
| Bostrom et al | 2022 Norway | To explore the experiences of people with chronic pain when engaging with the EPIO intervention program. | Digital self-directed pain-management program (EPIO). | Exploratory qualitative study | 15 adults (aged 29–74) living with chronic pain who had completed a feasibility pilot study of the EPIO digital pain self-management program. | Semi-structured interviews | Thematic analysis | • Engaging with EPIO • Coping with Pain in Everyday Life • The Value of engaging with the EPIO Program | High |

(*Continued*)

**Table 2.** (Continued)

| Authors | Year and Country | Aim of Study | Intervention format | Study Design | Participants | Data Collection | Data Analysis | Main Themes | Quality Appraisal (CASP) |
|---|---|---|---|---|---|---|---|---|---|
| Currie et al | 2015 Scotland | To examine attitudes towards current use of and acceptance of the use of technology in healthcare among older adults, aged 60–75, living with chronic pain. Aimed at 2 groups: The 'with technology' group had completed the intervention. The 'without technology' group had received regular home visits from health and /or social care professionals and did not use any digital pain management intervention. | Self-directed digital Cognitive Behavioural Therapy (CBT) pain App. | Mixed methods study: quantitative questionnaire and qualitative interviews. | 11 older adults (aged 60–75) living with chronic pain. The study interviewed a 'with technology group' (n = 4) who had completed Pathway through Pain, a self-directed online modular pain management programme, and a 'without technology group' (n = 7). | Semi-structured interviews. | Iterative Framework Approach | • Using technology to manage chronic pain<br>• Problems using technology in rural areas among patients with chronic pain | Low |
| Duggan et al | 2015 England | To explore participants' evaluations of SMART 2. | SMART2 –a digital pain mobile device App incorporating activity planning and review, feedback on behaviour and acceptance-based therapeutic exercises. | Qualitative study also using quantitative questionnaires. | 10 adults (mean age 50.3) with a history of chronic pain who had completed SMART 2 system. | Semi-structured interviews. | Thematic analysis | • Technical difficulties and beneficial effects<br>• Goal Setting<br>• Feedback<br>• Therapeutic content<br>• Process of behaviour change | Low |

**Table 2.** (Continued)

| Authors | Year and Country | Aim of Study | Intervention format | Study Design | Participants | Data Collection | Data Analysis | Main Themes | Quality Appraisal (CASP) |
|---|---|---|---|---|---|---|---|---|---|
| Geraghty et al | 2019 England | To explore patients' experiences of using an internet-based self-management support app for low back pain in primary care, with and without physiotherapist telephone guidance. | Support Back–a digital, internet-based self-management support app for low back pain. | Exploratory descriptive qualitative study, nested within a randomized feasibility trial. | 15 primary care patients over 18 years old with low back pain, who completed an online internet intervention ('SupportBack') with or without telephone support. | Semi-structured telephone interviews. | Thematic analysis | • Perceptions of Support Back's Design<br>• Engaging with the Support Back Intervention<br>• Promoting Positive thought processes<br>• Managing behaviour with 'Support Back'<br>• Feeling supported by physiotherapists on the telephone<br>• Severity of pain and comorbidity as barriers | High |
| Lawford et al | 2020 Australia | To qualitatively explore the perceptions and experiences of people with knee OA who participated in a randomised controlled trial involving an online PCST program. | 3-month digital pain management intervention (Pain Coping Skills Training (PCST)) for people with persistent knee pain. Participants also had 7 Skype consultation with a physiotherapist for prescription of an individualised, 3 times per week, home exercise program alongside the PCST intervention. | Qualitative study based on an interpretivist paradigm nested within a Randomised Controlled Trial (RCT). | 12 older adults (aged 50+) with persistent knee pain for more than 3 months who had completed a 3-month digital pain management intervention (Pain Coping Skills Training (PCST)) as part of a parallel RCT. | Semi-structured interviews. | Thematic analysis | • Easy to Understand and Follow<br>• Better able to cope with pain<br>• Anonymity and flexibility<br>• Not always relatable or engaging<br>• Support from clinician desirable | High |
| Nordin et al | 2017 Sweden | To explore patients' experiences of participation in a Web behaviour change programme. | Web behaviour change programme for activity in combination with multimodal rehabilitation for patients with persistent pain in primary health care. | Qualitative Interview Study. | 19 adult participants (mean age 45 years old) from a randomized controlled trial, who had completed an 8 module Web behaviour change programme for managing activity with chronic pain. | Semi-structured interviews. | Qualitative Content Analysis | • Take part in a flexible framework of own priority<br>• Acquire knowledge and insights<br>• Ways toward change<br>• Personal and environmental conditions influencing participation | High |

*(Continued)*

**Table 2.** (Continued)

| Authors | Year and Country | Aim of Study | Intervention format | Study Design | Participants | Data Collection | Data Analysis | Main Themes | Quality Appraisal (CASP) |
|---|---|---|---|---|---|---|---|---|---|
| Svendsen et al | 2022 Denmark | To qualitatively explore the implementation of selfBACK on (i) factors influencing embedding, integrating and sustaining engagement with the selfBACK app and (ii) the self-perceived effects, acceptability and satisfaction with the selfBACK app. | selfBACK–a 3 months digital pain management app for use on mobile phones. | Qualitative process evaluation alongside a Randomized Controlled Trial (RCT). | 25 adults (aged 21–78) with low back pain who had access to the selfBACK digital pain management app for 3 months and had completed the primary outcome questionnaires for a parallel RCT. | Semi-structured interviews. | Qualitative thematic analysis using a framework approach underpinned by Normalisation Process Theory (NPT). | • Level of embedding is associated with personal preferences, beliefs and level of information • Integration depends on perceived level of support and understanding of app features • Engagement depends on perceived fit of the App, time consumption, trustworthiness and functional issues • Perceived effects, acceptability, satisfaction, and sustained engagement and self-management | High |
| Wilson and Shaw | 2017 USA | To evaluate participants' perspectives after engaging in an digital pain self-management programme. | Digital Chronic Pain Management Program (CPMP)–an 8-week, self-directed, self-paced digital pain self-management programme. | Qualitative descriptive methodology | 55 adults (mean age 47 years) prescribed opioids for chronic pain who had completed the Chronic Pain Management Programme (CPMP). | Questionnaire—3 open ended survey questions. | Qualitative descriptive approach using content analysis. | • Positive reframing • Improved accountability • Feeling supported • Desire for personalising • Ease of use | High |
| Yu et al | 2020 Hong Kong | To develop and examine a digital pain management programme for people with ankylosing spondylitis. | 5-week digital pain management programme for people with ankylosing spondylitis incorporating mindfulness-informed exercises and CBT elements. | Mixed methods study using focus groups and quantitative pain outcome measures. | 30 participants (mean age 49 years) with Ankylosing Spondylitis from online Pain Management Programme (PMP). | Focus groups | Thematic analysis | • Better sleep quality • Environment limitations • Busy daytime schedule • Getting indolent with the exercises • Physical difficulties related to Ankylosing Spondylitis | Medium |

Interventions varied in the types of chronic pain that they addressed. One intervention specifically supported people with ankylosing spondylitis [22]. Two studies were aimed at chronic back pain [25, 26]. One study explored experiences amongst participants with arthritic pain [23], and one study explored experiences of participants with knee pain [27]. The other studies targeted patients with chronic pain but were not specific about the types of pain.

Studies also varied in their approaches to treatment for chronic pain, with over half focussing primarily on exercise and activity [21, 23, 25–28],and five studies were largely psychology-based interventions [9, 20, 22, 24, 29].

## Quality of included studies

Eight of the included studies were found to be of high quality after CASP analysis. Two studies [20, 21] were found to be low quality and one study [22] was found to be medium quality. To confirm robustness of this systematic review, the thematic analysis was repeated with the two low quality and one medium quality study excluded. The resultant themes and sub-themes were the same as in the original thematic analysis.

## Narrative synthesis

Three main themes were identified in the narrative synthesis which were mapped to the four constructs of NPT [12] to understand how individuals with chronic pain made sense of and implemented the intervention in their daily lives (see Fig 2). These themes and related sub-themes were: making sense of the digital intervention (tailoring to users' needs, human contact and support, accessibility of the digital intervention, personal and environmental factors affecting engagement with digital interventions); initiating and maintaining behaviour change (planning activity, being active); and personal growth (gaining understanding and skills, gaining and acting on feedback, negotiating a new relationship with pain).

## Making sense of the digital intervention

To make sense of the intervention and its relevance, individuals with chronic pain spoke about the importance of understanding what benefits the intervention offered, and how it would fit

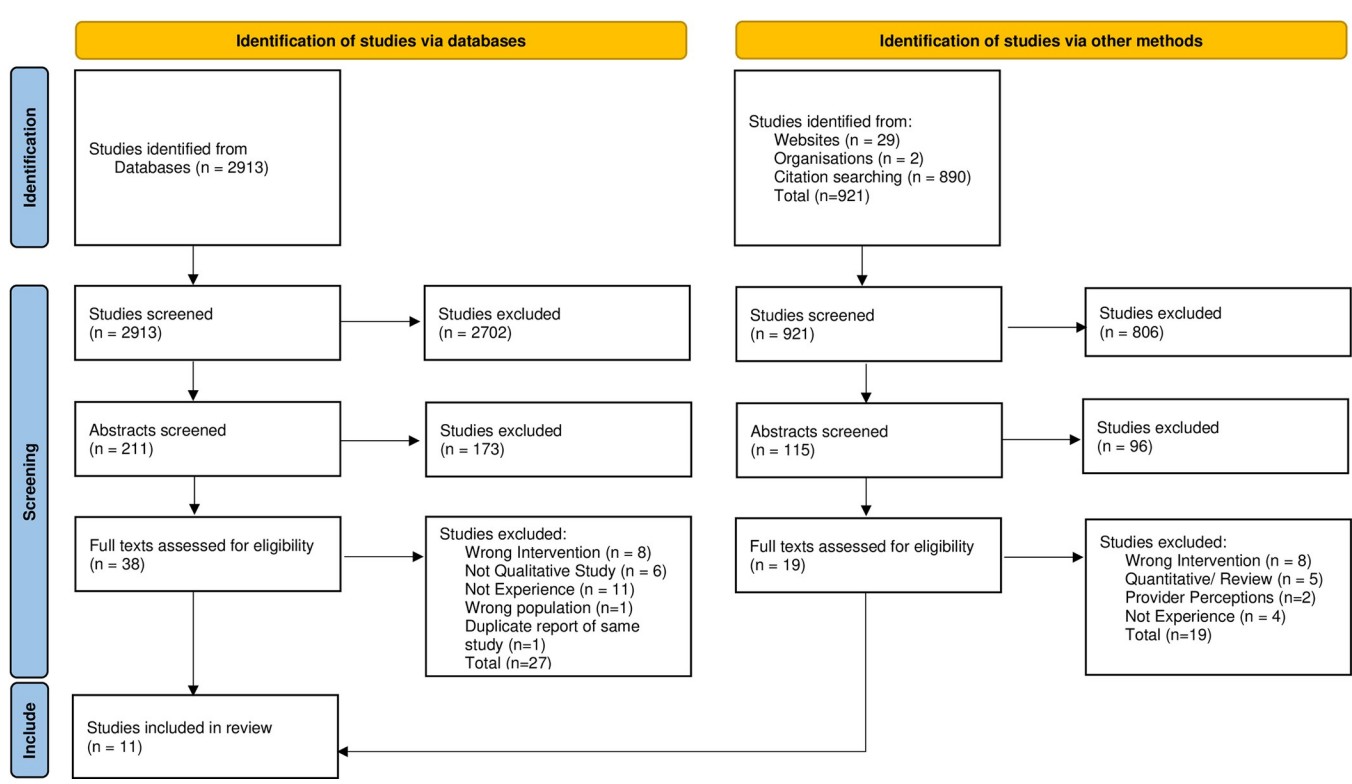

**Fig 1. PRISMA Flowchart of search results and included/ excluded studies.**

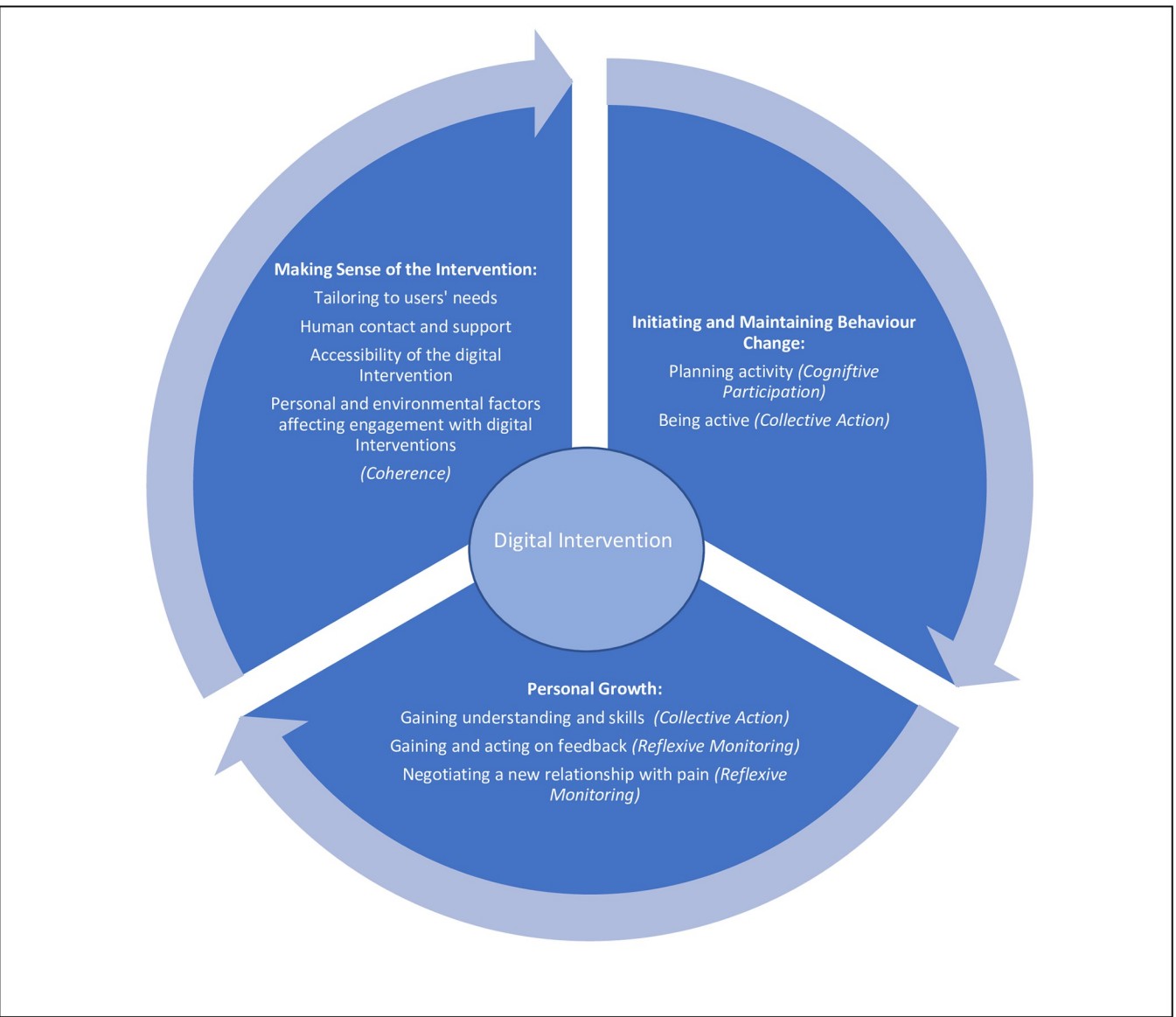

**Fig 2. Themes and related sub-themes in the narrative synthesis.**

into their lives, to perceive it as something useful and different from previous approaches. The theme aligns with the NPT construct *Coherence*, in which participants need to understand how the intervention differs from how they had previously managed their pain and to recognise its relevance to their everyday lives and needs, as well as building a shared understanding of how to implement the intervention with others, including health professionals [12, 16]. This theme comprised four subthemes reported below.

## Tailoring to users' needs

The NPT construct of coherence requires internalisation of the value of an intervention in order to engage with it and implement it in daily life [16]. Digital interventions that were personalised, flexible and specific to meet the individual needs of people with chronic pain

conditions and their lifestyles were felt to be more relevant [23, 26, 28, 29] and enabled users to develop confidence and trust in using the intervention [21, 23]. For example, in an Australian study, one woman with chronic arthritic pain commented:

> *". . . it can't just be, like you said, a generic thing. Because people aren't generic."* (Participant 11; Bhattarai et al, 2020, p3) [23].

Two studies reported that where participants had trust in the digital intervention and the information it provided. This was perceived to make the intervention more effective, reassuring individuals and helping them develop a greater level of self-comprehension, which enabled better recognition of their own behaviours and responses to pain [21, 28]. However, other users in three studies felt discouraged and frustrated where information provided by digital interventions was not new or relevant to supporting their problem [21, 29], or where deadlines for activities provided within the intervention made them feel pressurised or monitored to comply with the intervention [9].

## Accessibility of the digital intervention

Several studies found that the accessibility of digital interventions, including the simplicity of app design as well as the practical benefits of remote access enabled participants to make sense of, and engage with the interventions, allowing them to seem more usable and relevant, because they could fit self-management activities and exercises around their day-to-day lives [20, 23, 26, 29].

Several users across five studies reported that engaging with digital interventions was enabled by ensuring the digital intervention was easy to navigate [9, 21, 23, 26, 29]. However, disruption caused by technical difficulties, such as interrupted internet connectivity or system failures meaning that goal achievements were not recorded, were also considered to negatively impact on user experience and engagement [21, 29]. For instance, one participant in a UK study expressed disappointment that personal achievements related to self-management of the pain condition were not recorded due to technical problems within the app:

> *"If I looked back for a day when it hadn't connected it would come up with 4 red marks saying I hadn't done it and I'd find that really, really demoralising."* (Participant 8; Duggan et al, 2015, p57) [21].

Older users commented particularly on some of the physical difficulties using digital interventions, including visual and hearing difficulties, as well as difficulties with manual dexterity [20, 23]. Other users also reported that the amount of energy needed to engage in tasks to manage their chronic pain, or the prospect of long periods of computer-based work felt daunting and frustrating [9, 29]. Conversely, in another study which included a wider age-range of adults [24], respondents reported that easy accessibility to the app via a mobile phone was a positive motivating factor to engage with the intervention.

Engagement and accessibility among several participants in one study was perceived to be enhanced when clear explanations were provided using simple, non-clinical language, and where an audio-recorded voice was used the voice was seen as calming and clear [27]. Furthermore, individuals with chronic pain across three studies perceived that having more choice and a variety of content made the digital intervention more accessible and enabled them to choose specific therapeutic activities that addressed their own needs and priorities [20, 26, 28]. In two studies, some participants valued being able to complete therapeutic activities and exercises within the intervention at their own pace, which encouraged their engagement to continue to use it [20, 27].

## Human contact and support

Developing a sense of coherence around a particular digital intervention may require good communication between health care professionals, family and friends so that the person gained an understanding of how the intervention would work for them to help them manage their chronic pain in the context of their lives [16]. Individuals with chronic pain across nine studies found that contact with healthcare professionals, researchers or other participants, either alongside the digital pain management intervention or integrated within it, was generally positively received [9, 21, 23–27, 29]. In one study of a psychology-based digital self-management programme, participants reported that health care professional contact enabled people to ask questions in an empathetic environment so that they could better understand how to make the intervention work for them [9]. One participant this study commented on how therapeutic challenge by health professionals could help participants question their own beliefs and assumptions, allowing them to acknowledge their own thoughts and feelings in response to pain. This enabled a shift in focus from pain to engaging in everyday valued activities.

> *". . . when she questioned me back. . ., I had to think about pain in other ways. To not let it tear me down, rather put it aside, let it be, try not to focus on it, do something else, do something fun. . . That's what you need, a push, some back-up to get started when you're in pain all the time. . .."* (Participant 7; Bendelin et al, 2020, p5) [9].

Four studies that examined the perceived impact of digital interventions for the exercise management for chronic pain [24, 26–28] emphasised the beneficial role of contact with a healthcare professional, even if remotely, over only app delivered content, in enhancing engagement with the intervention. For example, in Geraghty et al's (2019) [26] UK study of an online exercise-based programme for chronic back pain, human therapy contact was perceived to help with motivation to exercise and made participants feel accountable for planned exercises.

Participants in two further studies also commented on the potential benefits of peer-to-peer engagement features within the intervention, to enable users to better support each other, decreasing a perceived sense of isolation [23, 29]. Several users in another American study of a CBT based App for opioid users with chronic pain, reported appreciating the opportunity to interact through chat groups with other people with similar chronic pain as they felt that other people in similar circumstances understood what they were going through, which helped reduce their sense of loneliness [29]. For example, one participant commented:

> *"I really enjoy being able to share my pain and experiences with others, it really helps knowing you're not alone. Before, I felt so alone that no one understood me."* (Wilson and Shaw, 2017, p3371) [29].

A number of participants across five studies reported that contact with therapists or other people living with chronic pain also helped them to overcome their own anxieties and feelings of loneliness by sharing their experiences and strategies for managing their pain condition [9, 20, 23, 24, 28]. This emphasis on sharing and understanding each other's experiences is consistent with the NPT concept of *Coherence*, in that that it relates to how a process of collective interpretation of a new practice or intervention with other people can help the individual to make sense and develop their understanding of a new practice and its value to them. However, conversely, in Lawford et al's (2020) [27] study of a digital pain coping skills training app for chronic knee pain, some participants felt more confident engaging with the application's virtual clinician, rather than with real people, as it gave them a sense of anonymity.

## Personal and environmental factors affecting engagement in digital interventions

In most studies individuals with chronic pain spoke about a range of personal and environmental factors considered to hinder their engagement with the digital pain intervention. This included: frustration with technical difficulties, infrastructure challenges such as slow broadband, lack of time and living space, distractions at home, level of severity of chronic pain, and apprehension about interacting with a digital application, all of which undermined trust and negatively affected participants' perception of the usefulness of the intervention to support them to manage their chronic pain [9, 20–23, 25, 26, 28, 29].

Age also affected older adults' experiences of engaging with, and accessing, digital pain management interventions, commonly due to both physical health difficulties and lack of confidence [20, 23]. In the two studies that specifically explored digital pain management interventions for older people, most participants reported difficulties in accessing internet interventions and applications in general due to a perceived lack of familiarity or proficiency with using the technology, or due to physical problems such as eyesight or manual dexterity [20, 23]. For example, one woman reported manual difficulties using a touch screen:

*"The slide was just difficult to move. . . If you move it to 'four' it just jumps back to 'three'. You had to be patient to get it to where you wanted it."* (Participant 17, Female, aged 75; Bhattarai et al, 2020, p4) [23].

Several individuals with chronic pain in one Swedish study of an Acceptance and Commitment Therapy-based digital pain management application [9], saw trust in the intervention as important for committing to implementing the intervention in their lives. This involved some acknowledgement that there was no medical solution to manage their chronic pain and that they had to be actively involved in their own care [9]. Being motivated and having previous positive experiences of exercise/ activity or therapeutic relationships with healthcare professionals were perceived as further factors further to facilitate greater engagement with self-directed digital interventions, with or without healthcare professional support [20, 25, 28]. For example, one woman with [type of chronic pain] using a digital behaviour change App commented:

*". . . I feel that one has to be motivated to participate in the course (the Web-BCPA) since it requires that I set aside time to log in to the program several times a day. . . it takes time to read all the texts and to do the assignments. . . "* (Interview 13, woman; Nordin et al, 2017, p6) [28].

## Initiating and maintaining behaviour change

Having made initial sense of the digital intervention, individuals across most studies strove to utilise new thinking patterns, practices, and changes to established behaviours within their daily lives [9, 21, 23–26, 28, 29]. Learning to utilise and adopt new patterns of behaviour involved both planning and implementing new practices, consistent with the NPT processes of *Cognitive Participation and Collective Action. Cognitive Participation* involves developing skills and planning how to adopt a new innovation, including engaging with others to achieve this. *Collective Action* involves developing and allocating resources and working with others to actually make changes in behaviours and practices to implement the new innovation into their everyday lives [12, 16]. This theme comprised two subthemes: planning activity and being active.

## Planning activity

Individuals across four studies reported needing to plan how to make the digital intervention work for them to support them with managing their chronic pain. By engaging with and participating in the digital intervention's activity planning, participants felt they played a more active role in their rehabilitation and became more aware of whether their patterns of activity were helpful or not, leading to meaningful behaviour change [9, 21, 26, 28]. Fitting the digital intervention around the stresses of everyday life was viewed as challenging for some users [24, 25]. However the digital intervention was also viewed by a number of other users as providing structure and routine, and the use of diarising functions, such as prompts and deadlines to support specific goals, helped to embed behaviour change and foster an increased sense of autonomy [9, 21, 23–26, 28]. For example, one participant in a study of a digital psychology-based pain management app commented that setting aside time in a stressful everyday life to read the app or practice exercises could be challenging, but the 'reminder function' was helpful:

> "... I easily forget to use it [the App] so reminders are nice, have to get it into everyday life." (Interview, Participant 34; Bostrom et al, 2022) [24].

Activity planning was reported by participants in three studies to provide them with the motivation to engage in specific therapeutic activities and increase physical activity, by involving them in setting manageable goals relevant to their own situation and lifestyle [21, 26, 28]. For example, one user of a digital activity-based App reported that the focus on planning their activities brought more structure to their daily lives:

> "It did make you plan for the future so you were thinking ahead... it added some structure to your day." (Participant 05; Duggan et al, 2015, p57) [21].

Participants developed new skills and different ways of interacting with the digital pain intervention consistent with the NPT construct of *Cognitive Participation*, which enabled them to break down goals into more manageable tasks, and to set and amend these goals. This helped them to identify goals that were viewed as relevant to them and their individual circumstances and provided pain self-management skills that they needed to develop for the future [21, 26, 28].

## Being active

The sub-theme of 'being active' links with the NPT construct of *Collective Action*. May et al (2007) [30] proposed that to move from planning and engagement with a new practice or innovation to actual implementation, people needed to build a pattern of dynamic interactions enabling them to work collectively with others to enact the digital intervention, to begin to integrate new behaviours into their everyday lives. In order to implement *Collective Action* [12, 16], individuals with health conditions need both to implement behavioural changes in their daily lives and to maintain these changes.

Individuals across several studies reported they needed to learn new approaches to adjust to new levels of activity and engage with the idea of increasing their physical function to manage their chronic pain [9, 21, 26, 28]. New approaches to managing pain encompassed the participants modifying their perception of pain, and working towards doing what was meaningful to them. For example, one person in a Swedish study highlighted:

> "A new attitude to life, I'd say. Stop thinking of what's coming next. Rather, do it now and do what you can...As when my son comes home from school telling me about his day–I really

*shouldn't plan dinner in my head meanwhile. Rather actually listen to him. That I learned during this treatment."* (Participant 1; Bendelin et al, 2020, p5) [9].

As participants started to integrate new behaviours into their daily lives, this broadened their scope of activity enabling them to achieve things they would not have previously attempted. Behaviour change was further reinforced by a sense of success in implementing the intervention's exercises and activities [21, 26]. Factors perceived among participants across two studies to facilitate adopting these new approaches included being motivated and interested in learning about pain management and open to trying something new [23, 28]. Use of a digital pain intervention was reported by a number of participants in over half of the studies to encourage people to add new forms of exercise into their daily routine and provide a structure for the maintenance of that routine by reinforcing regular practice and breaking down exercise into manageable amounts [21, 24–27, 29]. For example, one man in an exercise-based digital pain management app commented on how the app reinforced regular nightly exercise but enabled him to adjust the level to suit his needs:

*". . . . . . you could choose what exercises to do to help you, and you could also then either adjust the exercises up or down or stick where you are with them, as far as the number of types of movement you were doing each night. . ."* (Mike, 62, Internet Intervention, RMDQ = 2; Geraghty et al, 2020) [26].

## Personal growth

Over half of the included studies found that participants reported a sense of personal growth through engagement with the digital intervention, facilitating a more individualised understanding of their situation to make sense of their experience of pain and a greater awareness of the need for self-management of their chronic pain [9, 21, 24–28].

This theme comprised three subthemes: gaining understanding and skills; receiving and acting on feedback; and negotiating a new relationship with pain. The first sub-theme aligns with the NPT process of *Collective Action* as people learnt to implement the skills needed to manage their chronic pain. This enabled them to develop confidence to integrate changes and strategies in their lives and interact more effectively with health care professionals and others (such as family and friends) about their condition.

Following implementation and embedding of new practices, according to the NPT construct *Reflexive Monitoring* [12, 16], individuals are enabled to gather and evaluate feedback information about their experiences to further refine and change the way they do things in the future. By acting on feedback from digital pain interventions, individuals were able to then maintain behavioural changes and implement pain management strategies from the interventions into their daily lives. This ultimately led to a sense of personal growth enabling them to positively change the way they thought about and related to their pain [9, 21, 24–26, 28].

## Gaining understanding and skills

In four studies examining users' experiences of psychology-based digital pain interventions reported developing specific self-management skills, including distancing from the chronic pain, visualising their pain and relaxation techniques [9, 24, 28, 29]. This enabled them to notice nuances in their chronic pain condition and acquire a greater vocabulary regarding their condition with which to understand it themselves and to communicate with friends, family and healthcare professionals [9, 28]. For example, one person with chronic pain using an

Acceptance and Commitment Therapy (ACT) based digital pain management intervention found that visualising their pain as a physical object helped them not to blame themself for their condition:

> *"I had a grey lump. It was my pain and I could place it on the table and look at it. . . When blaming that thing instead of myself or something else. . . it was almost as if I was personalising pain. Because I made it into a thing, something concrete, as a grey substance".* (P7; Bendelin et al, 2020, p5) [9].

By developing further vocabulary regarding their chronic pain, individuals across two studies using a smartphone App with activity management and education reported they felt more confident in discussing their pain condition with health care professionals as well as with family and friends [23, 24, 28]. This enabled them to clarify issues of concern and more accurately inform professionals about changes in their condition.

Participants across five studies [9, 21, 24, 26, 28] reported gaining reassurance and confidence through increased self-knowledge from regularly using digital pain management interventions. This was because it helped them to normalise their own experiences of pain, understanding it as something that they shared with other people. Some individuals further reported that this reduced their feelings of fear when symptoms felt worse than usual and facilitated a sense of acceptance about their pain enabling them to develop more positive thought processes as coping strategies.

## Receiving and acting on feedback

The last two sub-themes within the personal growth theme broadly align to the NPT construct *Reflexive Monitoring* in which individuals with chronic pain reflected on the implementation of a new practice, developing plans for future adaptations and changes, and integrating these into their lives in a constant process of reconfiguration [16]. This sub-theme reflected how individuals became better able to monitor their chronic pain condition and evaluate the impact of new strategies or behaviours, facilitating further behaviour changes and developing their ability to adjust and adapt their responses to situations so as to better self-manage their pain.

Several users in four studies reported that they felt more confident to analyse and regularly review their behaviours, highlighting the centrality of self-management and leading to a greater awareness of the importance of setting, evaluating and analysing goals as an ongoing process, leading to increased behaviour awareness both in the present moment and at the time of reviewing [21, 26–28]. For example, one participant using a modular online behaviour change programme reported,

> *". . .before I started the assignment activity planning (in the Web-BCPA) I was not aware of how my behaviour related to the days with pain, but by monitoring this over time I started to plan my daily activities in a more balanced way. . ."* (P11; Nordin et al, 2017, p6) [28].

In five studies, digital pain management interventions which included feedback mechanisms to users about their daily activity levels enabled them to recognise when they were overdoing activity and this helped them to understand why they experienced their chronic pain, and led to reducing anxiety and distress associated with the pain [21, 26–29]. Furthermore, participants in three studies reported that such feedback also enabled them to develop a different mindset to activity, empowering them to make choices about the amount and type of daily

activity or exercise they did to manage their chronic pain, which contributed to their continued engagement with the intervention [21, 24, 25].

In four further studies a number of individuals also commented on how the use of reminders as a feedback tool within apps helped to embed a positive reframing of their chronic pain and sustain pain self-management strategies [21, 24, 26, 29]. This included regular reminders for users to check in on their thoughts and texts using positive affirmations [29]. For instance, one participant in an American study using digital pain management App for opioid users commented:

> "It was almost like having someone ask me not to think so negatively several times a day. That alone was very helpful, at least to me." (Wilson and Shaw, 2017, p59) [29].

### Negotiating a new relationship with pain

To embed new behaviours and a new approach to pain management in their lives individuals with chronic pain needed to negotiate a new relationship with pain, in which they reconciled themselves to living alongside their chronic pain acknowledging that this pain would not be cured [9, 24, 25]. Self-reflection and self-analysis facilitated by the digital pain management intervention [28] enabled participants to understand and re-define their old relationship with pain, allowing them to internalise the possibility of change [9, 21, 28]. For example, one participant in Bendelin et al's (2020) [9] study described how, by negotiating a new relationship with pain, they were able to reconnect with their old self:

> "Well it's been so long you tend to forget how it was and who you were. As for now, I'm mostly a pain patient. It's a good thing to get a feeling of and remembering how I was before and that I do exist even if I'm another person or something. I've still got my. . . not merely pain, but you know, I'm myself." (P3, Bendelin et al, 2020, p5) [9].

This negotiation of a new relationship with pain also required a shift in attitudes and behaviours related to pain, towards an acknowledgement of pain as being a part of the individual's everyday life [9, 24, 25, 29]. For example, one man in Svendsen et al's Denmark study using a digital intervention for people with chronic back pain commented:

> "I'm not feeling the same kind of pain anymore. I don't feel sorry for myself in the same way I used to" (P18; Svendsen et al, 2022, p7) [25].

Participants across five studies further reported how this enabled them to reframe their perception of pain to focus on more positive aspects of their lives [9, 24–26, 29]. For example, one participant in an American study using a digital pain management intervention for opioid users, described how they were able to shift away from negative thinking patterns that had been entrenched by the pain experience:

> "It was extremely helpful for me to shift my mind and spirit to focus on the good and wellness instead of sickness!" (Wilson and Shaw, 2017, p58) [29].

Several participants in five studies also commented that a new perceived relationship with pain meant accepting and coping with its presence in their everyday lives [9, 21, 24, 25, 29]. This enabled individuals to come to terms with the idea of living a life alongside pain, in which the presence of pain did not mean them having to put the pleasant aspects of life to one side

[21]. For example, one person in Bendelin et al's 2020 [9] study of a psychology-based digital pain management programme, using Acceptance and Commitment Therapy, summarised this acceptance of a closeness with pain:

> *"I'm in my pain in a different way now. I can't see it in front of me. I'm not trying to keep it at a distance. Rather I feel I'm with it most of the time."* (P9, Bendelin et al, 2020, p6) [9].

## Discussion

### Summary of main findings

This study highlighted the need for individuals with chronic pain participating in digital pain management interventions needed to value how the intervention was relevant to them and how it would fit into their lives. This enabled them to make sense of the intervention, in line with the NPT process of *Coherence*. To make sense of the digital intervention, it needed to be specific and tailored to the individual and to be presented in an accessible format, with sufficient relevant information. Personal motivation as well as the support of health care professionals facilitated their engagement, as did clear language, a wide variety of information and the facility to undertake the intervention at their own pace.

Participants described digital pain management interventions as providing structure, routine, accountability, and autonomy. This helped individuals to understand how the intervention could be integrated within their daily lives and make plans to embed new strategies, consistent with the NPT process of *Cognitive Participation*. Embedding new strategies and engaging with digital pain management interventions influenced behaviour change, (relating to the NPT process of *Collective Action*) by encouraging individuals to embed physical activity and exercise within their daily lives, to practice self-management activities, and to think about pain differently. This enabled participants to focus more on things they were able to achieve alongside their pain, and to work towards specific goals associated with meaningful activity.

Through engaging with digital interventions, people with chronic pain conditions gained new skills, increased confidence and reassurance about managing their condition, enabled also by sometimes by interacting with health care professionals, friends, and family members, further enhancing *Collective Action*. New skills learnt enabled them to evaluate and review their own attitudes, behaviours, and goals (*Reflexive Monitoring)* and to reframe their perception of pain in a more positive way. By doing so, they were able to redefine and renegotiate a new relationship to their chronic pain, with a reduced sense of fear and a greater acceptance of living alongside the pain in the context of their daily lives.

### How the findings relate to the existing literature

To our knowledge, no previous research has conducted a systematic review and narrative synthesis of the experiences of individuals with chronic pain participating in digital pain management interventions. A previous systematic review conducted across universities in Brazil, New Zealand and Australia [31] explored participants' experiences of telehealth for people with chronic pain in relation to barriers and facilitators to engagement. This study emphasised the importance of enabling individuals to work at their own pace, ensuring content was relevant, that interventions were flexible enough to fit into a patients' routine and highlighted technological challenges as a common barrier.

Gogovor et al's (2017) [32] review of existing content of internet-based chronic pain management interventions, involving focus group sessions with individuals with chronic pain, caregivers, and health professionals, highlighted the importance of interactive, personalised, tailored interventions with health professional contact to enhance self-management [32]. A

previous study [10] which explored experiences of digital interventions for hip and knee osteo-arthritis suggested that patients experience digital interventions for education and exercise as valid alternatives to traditional face to face healthcare treatment. Benefits included flexibility with regard to location and time. This study included regular and frequent contact with health care professionals, which was seen both as a necessity for a positive experience and also an advantage of this digital programme compared with traditional care. Cronström et al's (2019) [10] findings are broadly consistent with our research although it is not clear whether the experiences of digital interventions in our research's included studies were more or less positive than that of a traditional face-to-face healthcare intervention.

Conversely, Alley et al's (2019) [33] qualitative study of older adults' preferences for web-based physical activity found that many participants did not express an initial interest in web-based physical activity programs. In line with the findings from our study, these authors also reported that where participants engaged with the web-based physical activity programme, they preferred personalised interventions tailored to their needs. Participants also expressed a preference for simple interventions that were easy to navigate and enabled self-monitoring of their physical activity levels and progress towards goals that were relevant to their health condition.

Our study went beyond the existing literature, finding that healthcare professional support and previous positive experiences of therapeutic relationships with healthcare professionals could improve engagement with digital pain management interventions. Interactions with healthcare professionals and other users of the digital pain management intervention could reduce anxiety and loneliness, although some participants preferred the anonymity of engaging with a digital intervention rather than a real person. While older people found that physical difficulties (such as eyesight or manual dexterity problems) could be barriers to using digital pain management interventions, for wider users, environmental factors such as available space and busy lifestyles could also detrimentally affect engagement.

Our study also produced a novel finding regarding changing attitudes and behaviours and reframing perceptions of pain. Our research suggests that this reframing of perceptions requires commitment to implementing the intervention and leads to the negotiation of a new relationship with pain, enabling individuals to adapt to living a life alongside pain, in which they can identify and maintain valued activities rather than putting the pleasant aspects of life to one side. The ability to maintain this focus on valued living involves a continual process of acknowledging, reviewing, and adapting their own attitudes, behaviours and responses to accommodate changes in the chronic pain condition and other life changes.

## Strengths and limitations of the synthesis

We investigated studies exploring the experiences of people engaging in digital interventions for chronic pain management support. The majority of studies [9, 20–23, 26, 28, 29] were conducted prior to the Covid 19 pandemic and therefore do not reflect the impact of the pandemic on participants' experiences. However, three studies [24, 25, 27] were conducted during or after Covid-19 and reflect the experiences of participants in digital pain management interventions throughout this period.

Methodologically, our study used an interpretive stance which was appropriate to explore the lived experience of individuals with chronic pain using digital health interventions. The use of NPT [12] facilitated the development of theoretical and empirical insights into the processes by which people engaged with digital pain management interventions and how they subsequently embed new practices and behaviours into their daily life, as well as those practices to enable sustained behaviour change and personal growth in the future.

The Critical Appraisal Skills Programme (CASP) Qualitative Research Checklist [17] was used to provide a framework for appraising the quality of the included studies. A consensus view of the quality of included studies from the initial database search (11/06/21) was achieved by the use of two researchers (JS and JH) independently reviewing and then discussing studies. This was not possible for the second database search (07/07/23) and the studies were reviewed and appraised for quality by JS alone.

To reduce bias, it would have been preferable for more than one author to independently fully analyse each paper and to identify themes and sub-themes. As well as improving confidence in the findings, this may have yielded other insights into participants' experiences. However, this was not possible and the full analysis of included studies was conducted by JS alone. To confirm robustness of the systematic review, the thematic analysis was repeated with the three low and medium quality studies removed. All originally identified themes and sub-themes were present across the included studies, suggesting robustness of the narrative synthesis.

The identified themes and sub-themes in the narrative synthesis were discussed with two other authors in the team (ES, LW), one with qualitative research expertise (ES) to reach a consensus view on final themes and sub-themes.

## Conclusion and implications for practice, policy and future research

Our study provides useful insights into peoples' experiences of digital healthcare interventions to support their management of chronic pain, which could help inform the development of such interventions in a post-Covid 19 healthcare community. It has highlighted several features of digital interventions, such as the need for specific and tailored content, a focus on changing attitudes and behaviours, and a structured framework for goal setting, which would be likely to make the digital intervention more beneficial and effective in supporting self-management for individuals living with chronic pain. The key recommendations from our findings are that digital pain management interventions should:

- Provide specific and tailored information for individual participants.

- Provide a focus on changing attitudes and behaviours and reframing perceptions of pain.

- Provide structured goal setting with prompts to review goals.

- Consider providing health care professional support integrated alongside the digital intervention.

The development of digital interventions and resources to support patient care is a key element of NHS England's vision for a digital future to deliver personalised care more effectively and efficiently, to improve long-term sustainability of health and social care services [34]. There is growing recognition of the importance of supported self-management for people with chronic pain, using self-management interventions which enhance patient self-efficacy and empower patients and clinicians to play a shared role in decision making and treatment [35]. Digital interventions are well-placed to facilitate supported self-management, as they can support patients between contacts with healthcare professionals, enabling patients to progress their management plan independently, and may provide an ongoing support resource long after an episode of care has finished.

Given the limited literature on this topic, further qualitative studies regarding patients' experiences of digital pain management interventions will be beneficial and may yield insights into the specific requirements of digital interventions, as distinct from face-to-face interventions. In particular, further research needs to focus on the extent to which social or

demographic factors influence peoples' experiences of digital pain management interventions. It would also be useful to explore the experiences of people who disengage from digital pain management interventions to understand their experience of these interventions and what led them to disengage. This may help to improve the accessibility and user experience of future digital interventions, and potentially result in improved outcomes from these interventions. By more fully understanding the impact of these interventions on different groups of people, we will be able to develop future digital pain management interventions with the specific needs of different groups of service users in mind (Bhattarai et al, 2020). This could include tailoring digital interventions to accommodate individuals' personal learning needs, preferences and differing levels of confidence with digital media. Digital interventions may also be developed to provide specialist support to people with complex multiple comorbidities, of which pain is a significant factor, such as Long Covid, diabetes and other long-term conditions. To achieve this, we will also need increased service-user/ patient and public involvement at all stages of the design, implementation and evaluation of digital interventions, and co-ordination between different clinical specialisations and organisations within healthcare services.

## Supporting information

**S1 Checklist. PRISMA checklist.**
(PDF)

## Acknowledgments

The authors would like to acknowledge the contribution of Dr Jane Hazelgrove (Consultant in Pain Medicine, Southern Health NHS Foundation Trust) to this research.

## Author Contributions

**Conceptualization:** Justin Damian Russell Strain, Lindsay Welch, Euan Sadler.

**Data curation:** Justin Damian Russell Strain, Lindsay Welch, Euan Sadler.

**Formal analysis:** Justin Damian Russell Strain, Lindsay Welch, Euan Sadler.

**Funding acquisition:** Justin Damian Russell Strain, Euan Sadler.

**Investigation:** Justin Damian Russell Strain.

**Methodology:** Justin Damian Russell Strain, Lindsay Welch, Euan Sadler.

**Project administration:** Justin Damian Russell Strain, Euan Sadler.

**Visualization:** Justin Damian Russell Strain, Lindsay Welch, Euan Sadler.

**Writing – original draft:** Justin Damian Russell Strain.

**Writing – review & editing:** Lindsay Welch, Euan Sadler.

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
