## [Decision Letter · Decision Letter 0]

6 May 2024

PONE-D-24-09664Systematic Review and Narrative Synthesis of the experiences of individuals with chronic pain participating in digital pain management interventions.PLOS ONE

Dear Dr. Strain,

Thank you for submitting your manuscript to PLOS ONE. After careful consideration, we feel that it has merit but does not fully meet PLOS ONE’s publication criteria as it currently stands. Therefore, we invite you to submit a revised version of the manuscript that addresses the points raised during the review process.

We look forward to receiving your revised manuscript.

Kind regards,

Mehrnaz Kajbafvala, Ph.D

Academic Editor

PLOS ONE

“JS received a Research Internship and Research Initiation Award to provide funded time for this research.  This included academic supervision from ES and LW.  These awards were both funded by NIHR Applied Research Collaboration (ARC) Wessex (https://www.arc-wx.nihr.ac.uk/) and NHS England (https://www.england.nhs.uk/).”

Reviewers' comments:

Reviewer's Responses to Questions

**Comments to the Author**

1. Is the manuscript technically sound, and do the data support the conclusions?

Reviewer #1: Yes

Reviewer #2: Yes

2. Has the statistical analysis been performed appropriately and rigorously? 

Reviewer #1: Yes

Reviewer #2: Yes

3. Have the authors made all data underlying the findings in their manuscript fully available?

Reviewer #1: Yes

Reviewer #2: Yes

4. Is the manuscript presented in an intelligible fashion and written in standard English?

Reviewer #1: Yes

Reviewer #2: Yes

5. Review Comments to the Author

Reviewer #1: The aim of this study was to systematically review and synthesise evidence from qualitative studies regarding the experiences of individuals with chronic pain participating in digital pain management interventions in primary care and community settings. The key recommendations from this study are that digital pain management

interventions should provide:

Specific and tailored information for individual participants.

Focus on changing attitudes and behaviours and reframing perceptions of pain.

Structured goal setting with prompts to review goals.

Potential healthcare professional support alongside the digital intervention.

Different items of this Manuscript is very well written and is appropriate for publication in this Journal.

Reviewer #2: Dear Authors,

Thank you very much for your interesting article that discusses an important topic. The title of your article " Systematic Review and Narrative Synthesis of the experiences of individuals with chronic pain participating in digital pain management interventions " is an interesting idea and I enjoyed reading it. However, I have several suggestions for improving your article.

FIRST; Please ensure that your manuscript meets PLOS ONE's style requirements, including those for file naming. The PLOS ONE style templates can be found at

https://journals.plos.org/plosone/s/file?id=ba62/PLOSOne_formatting_sample_title_authorsaffiliations.pdf

Abstract

The abstract section should be written according to the guidelines of the journal (Background, Methods, Results, Conclusion).

Please write keywords based on MeSH terms.

Methodology

In the " search strategy " section, Google Scholar is deleted; Because it is not a database but a search engine.

Discussion

I suggest you explain more how your study will help the health system and patients.

6. PLOS authors have the option to publish the peer review history of their article (what does this mean?). If published, this will include your full peer review and any attached files.

Reviewer #1: No

Reviewer #2: **Yes: **Arsalan Ghorbanpour

---

## [Author Response · Author response to Decision Letter 0]

7 Jun 2024

Dr M Kajbafvala 

Academic Editor

PLOS ONE 7th June 2024

Dear Dr Kajbafvala,

Re our manuscript: "Systematic Review and Narrative Synthesis of the experiences of individuals with chronic pain participating in digital pain management interventions".

Thank you for giving us the opportunity to revise our manuscript and we are pleased that you found our article interesting. Please see our following responses to your reviewer’s comments outlining how we have addressed the several suggestions you have indicated for further improving our article.

https://journals.plos.org/plosone/s/file?id=ba62/PLOSOne_formatting_sample_title_authorsaffiliations.pdf

File naming style requirements have now been met – this is only applicable to Fig1 and Fig2. 

2. Funding Statement: Thank you for stating the following financial disclosure:“JS received a Research Internship and Research Initiation Award to provide funded time for this research. This included academic supervision from ES and LW. These awards were both funded by NIHR Applied Research Collaboration (ARC) Wessex (https://www.arc-wx.nihr.ac.uk/) and NHS England (https://www.england.nhs.uk/). ”Please state what role the funders took in the study. If the funders had no role, please state: "The funders had no role in study design, data collection and analysis, decision to publish, or preparation of the manuscript." If this statement is not correct you must amend it as needed. Please include this amended Role of Funder statement in your cover letter; we will change the online submission form on your behalf.

We have added the following to the funding statement: ‘ES is supported by NIHR ARC Wessex’. I confirm that the funders had no role in study design, data collection and analysis, decision to publish, or preparation of the manuscript.

3. Data Availability Statement: Please confirm at this time whether or not your submission contains all raw data required to replicate the results of your study. 

I confirm that all relevant data are within the manuscript. 

4. Supporting Files: Please include captions for your Supporting Information files at the end of your manuscript.

Other than 2 figures which appear within the main manuscript (Fig1 and Fig2) there are no additional Supporting Files. Fig1 and Fig2 have been uploaded to PACE and have met PLOS requirements.

5. Reference List: Please review your reference list to ensure that it is complete and correct.

References have been rewritten as numbered references to comply with the journal’s style requirements.

6. Abstract: The abstract section should be written according to the guidelines of the journal (Background, Methods, Results, Conclusion). Please write keywords based on MeSH terms.

The abstract has now been written in accordance with the journal guidelines. MeSH terms: Pain Management, Chronic Pain and Qualitative Research have also been added.

7. Methodology: In the " search strategy " section, Google Scholar is deleted; Because it is not a database but a search engine.

Google Scholar has now been deleted from the search strategy section but kept in description of citation tracking.

8. Discussion: I suggest you explain more how your study will help the health system and patients.

We thank the editor/reviewer for this comment. We have now added further detail to the discussion on how our study will help the health system and patients.

Thank you again for giving us the opportunity to revise our manuscript. We look forward to hearing back from you at your earliest convenience. 

Yours sincerely. 

Justin Strain, and on behalf of my other authors. 

Justin Strain (He/ Him)

Consultant Musculoskeletal Allied Health Professional

Clinical Specialist Physiotherapist in Pain Management and Spines

Southern Health NHS Foundation Trust

Justin.Strain@southernhealth.nhs.uk

---

## [Decision Letter · Decision Letter 1]

19 Jun 2024

Systematic Review and Narrative Synthesis of the experiences of individuals with chronic pain participating in digital pain management interventions.

PONE-D-24-09664R1

Dear Dr. Justin Damian Russell Strain

We’re pleased to inform you that your manuscript has been judged scientifically suitable for publication and will be formally accepted for publication once it meets all outstanding technical requirements.

Kind regards,

Mehrnaz Kajbafvala, Ph.D

Academic Editor

PLOS ONE

Additional Editor Comments (optional):

Reviewers' comments:

Reviewer's Responses to Questions

**Comments to the Author**

1. If the authors have adequately addressed your comments raised in a previous round of review and you feel that this manuscript is now acceptable for publication, you may indicate that here to bypass the “Comments to the Author” section, enter your conflict of interest statement in the “Confidential to Editor” section, and submit your "Accept" recommendation.

Reviewer #2: All comments have been addressed

Reviewer #3: All comments have been addressed

2. Is the manuscript technically sound, and do the data support the conclusions?

Reviewer #2: Yes

Reviewer #3: Yes

3. Has the statistical analysis been performed appropriately and rigorously? 

Reviewer #2: N/A

Reviewer #3: Yes

4. Have the authors made all data underlying the findings in their manuscript fully available?

Reviewer #2: Yes

Reviewer #3: Yes

5. Is the manuscript presented in an intelligible fashion and written in standard English?

Reviewer #2: Yes

Reviewer #3: Yes

6. Review Comments to the Author

Reviewer #2: (No Response)

Reviewer #3: All comments have been addressed adequately. The manuscript in this present form is acceptable for publication.

7. PLOS authors have the option to publish the peer review history of their article (what does this mean?). If published, this will include your full peer review and any attached files.

Reviewer #2: **Yes: **Arsalan Ghorbanpour

Reviewer #3: No

---

## [Editor Report · Acceptance letter]

27 Jun 2024

PONE-D-24-09664R1 

PLOS ONE

Dear Dr. Strain, 

I'm pleased to inform you that your manuscript has been deemed suitable for publication in PLOS ONE. Congratulations! Your manuscript is now being handed over to our production team.

Kind regards, 

on behalf of

Dr. Mehrnaz Kajbafvala 

Academic Editor

PLOS ONE